# Leucine Supplementation Prevents the Development of Skeletal Muscle Dysfunction in a Rat Model of HFpEF

**DOI:** 10.3390/cells13060502

**Published:** 2024-03-13

**Authors:** Paula Ketilly Nascimento Alves, Antje Schauer, Antje Augstein, Maria-Elisa Prieto Jarabo, Anita Männel, Peggy Barthel, Beatrice Vahle, Anselmo S. Moriscot, Axel Linke, Volker Adams

**Affiliations:** 1Heart Center Dresden, Laboratory of Molecular and Experimental Cardiology, TU Dresden, 01307 Dresden, Germany; paulaketilly@usp.br (P.K.N.A.); antje.schauer@tu-dresden.de (A.S.); antje.augstein@tu-dresden.de (A.A.); maria-elisa.prieto@uniklinikum-dresden.de (M.-E.P.J.); anita.maennel@tu-dresden.de (A.M.); beatrice.vahle@tu-dresden.de (B.V.); axel.linke@tu-dresden.de (A.L.); 2Department of Anatomy, Institute of Biomedical Sciences, University of Sao Paulo, São Paulo 05508000, Brazil; moriscotanselmo@gmail.com

**Keywords:** HFpEF, ZSF1, leucine, mitochondria, diastolic dysfunction, skeletal muscle dysfunction

## Abstract

Heart failure with preserved ejection fraction (HFpEF) is associated with exercise intolerance due to alterations in the skeletal muscle (SKM). Leucine supplementation is known to alter the anabolic/catabolic balance and to improve mitochondrial function. Thus, we investigated the effect of leucine supplementation in both a primary and a secondary prevention approach on SKM function and factors modulating muscle function in an established HFpEF rat model. Female ZSF1 obese rats were randomized to an untreated, a primary prevention, and a secondary prevention group. For primary prevention, leucine supplementation was started before the onset of HFpEF (8 weeks of age) and for secondary prevention, leucine supplementation was started after the onset of HFpEF (20 weeks of age). SKM function was assessed at an age of 32 weeks, and SKM tissue was collected for the assessment of mitochondrial function and histological and molecular analyses. Leucine supplementation prevented the development of SKM dysfunction whereas it could not reverse it. In the primary prevention group, mitochondrial function improved and higher expressions of mitofilin, Mfn-2, Fis1, and miCK were evident in SKM. The expression of UCP3 was reduced whereas the mitochondrial content and markers for catabolism (MuRF1, MAFBx), muscle cross-sectional area, and SKM mass did not change. Our data show that leucine supplementation prevented the development of skeletal muscle dysfunction in a rat model of HFpEF, which may be mediated by improving mitochondrial function through modulating energy transfer.

## 1. Introduction

Heart failure (HF) with preserved ejection fraction (HFpEF) accounts for approximately 50% of all HF cases, but over the next decade, it will become the more dominant form [1]. This expected increase is due to an increase in life expectancy and an increasing prevalence of comorbidities such as metabolic syndrome, obesity, and diabetes mellitus. 

Besides structural alterations in the myocardium like stiffening of the ventricle [2], increased fibrosis [3], and functional impairment, HFpEF is also associated with endothelial [4,5] and skeletal muscle (SKM) dysfunction/atrophy [6,7,8,9]. These skeletal muscle alterations are at least partially responsible for exercise intolerance and early fatigue, hallmarks of HFpEF patients. The abnormalities observed in the SKM of HFpEF patients or experimental models of the disease include the following: decreased muscle mass [8,10]; a reduced type I-to-type-II fiber ratio as well as a reduced capillary-to-fiber ratio [7,11,12]; abnormal fat infiltration into the thigh SKM [13,14]; increased levels of atrophy genes and proteins [8,15]; reduction in mitochondrial function and content [16,17,18]; and rapid depletion of high-energy phosphate during exercise with markedly delayed repletion of high-energy phosphate during recovery [19]. Since exercise intolerance is associated with reduced quality of life and increased mortality [20,21,22], the search for therapeutic options to improve exercise capacity and the translation to the clinical scenario is performed with great effort. So far, exercise training [23,24] and sodium–glucose cotransporter 2 inhibitors (SGLTi) [25] have demonstrated positive effects with respect to improved quality of life and reduced mortality. Furthermore, exercise training [11,26] and SGLT2i [9] exert positive effects on SKM function and structural/molecular alterations. At the preclinical level, the inhibition of the atrophy-inducing protein MuRF1 also seems to improve muscle performance in HFpEF [8].

Leucine is an essential, anabolic, branched-chain amino acid that promotes muscle protein synthesis by activating the mechanistic target of rapamycin (mTOR) [27]. Leucine supplementation has been intensively studied in humans, and varying results have been reported [28,29,30,31]. Regarding experimental models, a leucine-rich diet has been reported to attenuate SKM loss [32] probably by modulating protein breakdown via the ubiquitin–proteasome system [33]. Furthermore, our group recently demonstrated that leucine reduced muscle atrophy and muscle dysfunction in a hindlimb immobilization model [34]. In addition, leucine improved myocardial diastolic function in an HFpEF animal model [35]. Besides modulating protein degradation pathways, a beneficial effect of leucine on mitochondrial function (energy generation, proteins of the tricarboxylic cycle) has been reported [35,36]. Since skeletal muscle atrophy/dysfunction in HFpEF is also at least partially due to the dysregulation of anabolism and catabolism and to dysfunctional mitochondria [15], the aim of the present study was to evaluate whether a therapeutic approach via a leucine-rich diet is beneficial in HFpEF with respect to skeletal muscle function.

## 2. Materials and Methods

### 2.1. Study Design

In total, 36 female ZSF1 obese rats (Charles River Laboratories, Sulzfeld, Germany) were used in this study. The animals were randomized into three groups: HFpEF (n = 12, animals were fed standard chow throughout the experiment); HFpEF-prim (n = 12, primary prevention group, animals were fed with standard chow enriched with 3% leucine starting at an age of 8 weeks); and HFpEF-sec (n = 12, secondary prevention group, animals were fed with standard chow enriched with 3% leucine starting at an age of 20 weeks, when HFpEF had already developed) (Figure 1). All animals were maintained under regular conditions: controlled temperature (24 ± 1 °C, 12 h light–dark cycle), food and water provided ad libitum. The leucine concentration of 3% was chosen based on earlier studies by our group [34,35]. The time points for initiating the therapy with leucine at 8 and 20 weeks were chosen based on earlier studies showing that at 8 weeks no signs of HFpEF or skeletal muscle dysfunction were seen, whereas at 20 weeks of age, muscle dysfunction and diastolic dysfunction with preserved ejection fraction were evident [37]. 

At an age of 32 weeks, echocardiography was performed and the animals were sacrificed (Figure 1). The right soleus muscle (SO) was prepared for functional measurements, and a small piece of the left SO was used for the preparation of saponin skinned muscle fibers for the measurement of mitochondrial respiration. In addition, SKM weights (SO and EDL) were determined and SKM tissue was frozen in liquid nitrogen for subsequent molecular analyses or fixed in PBS buffered formalin for histological analyses.

For the assessment of muscle force and mitochondrial respiration, a control group (ZSF1-lean animals, 32 weeks of age, n = 9) was included in the analyses. 

This study was approved by the local animal research council, TU Dresden, and the Landesbehörde Sachsen (TVV 26/2022).

### 2.2. Echocardiography

To assess cardiac function, the rats were anesthetized by isoflurane (1.5–2%), and transthoracic echocardiography was performed using a Vevo 3100 system and a 21-MHz transducer (FUJIFILM Visual Sonics, Amsterdam, Netherlands) [37,38]. For systolic function, the B- and M-Mode of the parasternal long- and short-axis were measured at the level of the papillary muscles. Diastolic function was assessed in the apical 4-chamber view using pulse wave Doppler (to assess early (E) and atrial (A) waves of the mitral valve velocity) and tissue Doppler (for quantification of myocardial velocity (e’)) at the level of the basal septal segment in the septal wall of the left ventricle. To obtain functional parameters (i.e., LV ejection fraction (LVEF) and stroke volume (SV) and E/e’ ratio), the Vevo LAB 3.1.1 software (FUJIFILM Visual Sonics, Amsterdam, Netherlands) was used.

### 2.3. Skeletal Muscle Function

The right soleus was prepared and attached vertically between a hook and force transducer in a Krebs–Henseleit buffer-filled organ bath (1205A: Isolated Muscle System—Rat, Aurora Scientific Inc., Aurora, ON, Canada). To assess ex vivo muscle function, platinum electrodes were used to stimulate muscle contraction with a supra-maximal current (700 mA, 500 ms train duration, 0.25 ms pulse width) using a high-power bipolar stimulator (701C; Aurora Scientific Inc., ON, Canada). The attached muscle was set at an optimal length (Lo) (length where the maximal twitch force was generated). Thereafter, a force–frequency protocol was performed (1, 15, 30, 50, 80, 120, and 150 Hz), separated by 1-minute rest intervals.

After a 5-minute rest-period, during which the muscle length was measured, the muscle was stimulated every 2 s with 40 Hz over 5 min to assess muscle fatigue. The muscle was subsequently detached, trimmed free from fat and tendon, blotted dry on filter paper, and weighed. Muscle force (N) was normalized to muscle cross-sectional area (cm^2^) by dividing muscle mass (g) by the product of optimal length (cm) and estimated muscle density (1.06 g/cm^3^), which allowed specific force (N/cm^2^) to be calculated [37,38,39].

### 2.4. Assessment of Mitochondrial Function

Saponin-skinned muscle fibers were used to measure mitochondrial function as recently described [35]. Oxygen consumption rates were determined by using a Clark electrode (Strathkelvin Instruments, Motherwell, UK) in an oxygraphic cell at 25 °C with continuous stirring. To avoid oxygen diffusion limitations, oxygen concentration in the cell was increased above 400 µmol/L by adding pure oxygen. SO fibers were isolated in permeabilization solution (SolP) containing the following (in mmol/L): 2.77 CaK_2_EGTA, 7.23 K_2_EGTA, 6.56 MgCl_2_, 5.7 Na_2_ATP, 15 phosphocreatine (PCr), 20 taurine, 0.5 DTT, 50 K-methane-sulfonate, 20 imidazole (pH 7.1). The fibers were then incubated for 30 min in SolP with 50 μg/mL saponin. After this incubation period, the fibers were transferred to a respiration solution (SolR) containing (in mmol/L) 20 taurine, 20 HEPES, 10 KH_2_PO4, 0.5 EGTA, 3 MgCl_2_, 0.11 sucrose, 60 K-lactobionate (pH 7.4) for 10 min to remove adenine nucleotides and PCr. For the measurements of mitochondrial respiration rates, 1–5 mg of skinned fibers were added to the oxygraphic cell containing 1 mL of SolR supplemented with 1 mg/ml bovine serum albumin. The following substrates were added sequentially and oxygen consumption was monitored continuously: (i) glutamate (10 mmol/L), malate (2.0 mmol/L), (complex I state 2 respiration); (ii) adenosine diphosphate (5.0 mmol/L; complex I state 3 respiration); (iii) octanoylcarnitin (0.2 mmol/L; complex I state 3); (iv) cytochrome C (10 μmol/L; test for membrane integrity); (v) succinate (10 mmol/L; oxidative phosphorylation of complex I + II); (vi) rotenone (0.5 mmol/L; oxidative phosphorylation of complex II); (vii) FCCP (0.5 μmol/L; maximal uncoupled complex II respiration); (viii) antimycin A (2.5 μmol/L, as a complex III inhibitor); (ix) ascorbate/N, N, N′, N′-tetramethyl-p-phenylenediamine dichloride (2 mmol/L/0.5 mmol/L, maximal uncoupled complex IV respiration). After the experiment, the fiber bundles were blotted dry and weighed. Rates of respiration are given in nmoles O_2_ per second per mg wet weight.

### 2.5. Western Blot Analyses 

Frozen muscle samples were homogenized in Relax buffer (90 mmol/L HEPES, 126 mmol/L KCl, 36 mmol/L NaCl, 1 mmol/L MgCl, 50 mmol/L EGTA, 8 mmol/L ATP, 10 mmol/L Creatinphosphate; pH 7.4) containing a protease inhibitor mix (Inhibitor mix M, Serva, Heidelberg, Germany). The BCA assay (Pierce, Bonn, Germany) was used to measure protein concentration and the aliquots (5–20 μg) were separated by SDS-polyacrylamide gel electrophoresis. After transfer of the proteins to a polyvinylidene fluoride membrane (PVDF), the PVDF membrane was blocked with 5% milk and incubated overnight at 4 °C using the following primary antibodies: MafBx, (1:1000, Abcam, Cambridge, UK); total OXPHOS components (1:250, Abcam, Cambridge, UK); Fis1, porin (VDAC1), and mi-CK (all 1:1000, Proteintech, Planegg-Martinsried, Germany); MCU (1:1000, CellSignaling, Danvers, MA, USA); UCP3 (1:1000, Thermo Fisher, Waltham, MA, USA); mitofilin and MuRF1 (all 1:200, Santa Cruz, Heidelberg, Germany). After a short wash, the membranes were incubated with a horseradish peroxidase-conjugated secondary antibody and specific bands were visualized by chemiluminescence (Super Signal West Pico, Thermo Fisher Scientific Inc., Bonn, Germany). To quantify protein expression, densitometry was performed using the 1D scan software package version 15.08b (Scanalytics Inc., Rockville, MD, USA), and measurements were normalized to the loading control GAPDH (1/5000; HyTest Ltd., Turku, Finland). Full western blot pictures are depicted in the Appendix A. 

### 2.6. Citrate Synthase Activity Measurement

The SO muscle was homogenized in Relax buffer and the aliquots were used for enzyme activity measurements. The enzyme activity for citrate synthase (CS, EC 2.3.3.1) was measured by spectrophotometry [40]. Enzyme activities are expressed as mU/mg protein. 

### 2.7. Statistical Analyses

Data are presented as mean ± SEM. One-way analysis of variance (ANOVA) followed by a post hoc (Newman–Keuls Multiple Comparison Test) was used to compare groups, while two-way repeated measures ANOVA followed by Tukey’s multiple comparison test was used to assess contractile function (GraphPad Prism 7.0). Significance was accepted as *p* < 0.05.

## 3. Results

### 3.1. Animal Characteristics

Assessment of diastolic function by echocardiography at an age of 20 weeks revealed a significantly higher E/é ratio in HFpEF and HFpEF-sec compared to HFpEF-prim, while no difference was seen between HFpEF and HFpEF-sec (HFpEF: 24.25 ± 0.59; HFpEF-prim: 18.39 ± 0.68 ***^,###^; HFpEF-sec: 24.25 ± 1.15; *** *p* < 0.001 vs. HFpEF, ^###^ *p* < 0.001 vs. HFpEF-sec). Comparing the E/é ratio to the mean value of age-matched ZSF-1 lean healthy control animals (17.80 ± 0.85) indicates that HFpEF- and HFpEF-sec animals at 20 weeks of age exhibit pronounced signs of diastolic dysfunction with still preserved left ventricular ejection fraction (HFpEF: 66.7 ± 1.1%; HFpEF-prim: 68.5 ± 1.3%; HFpEF-sec: 70.5 ± 2.1%). At the end of the study (32 wks of age), E/é was significantly lower in HFpEF-prim and HFpEF-sec compared to untreated HFpEF (HFpEF: 24.42 ± 0.41; HFpEF-prim: 18.46 ± 0.53 ***; HFpEF-sec: 18.68 ± 1.05 ***; *** *p* < 0.001 vs. HFpEF) while LVEF was preserved (HFpEF: 71.7 ± 1.2%; HFpEF-prim: 68.1 ± 1.3%; HFpEF-sec: 71.1 ± 1.0%). In summary, the animals randomized into HFpEF-prim and HFpEF-sec displayed signs of HFpEF at the time of randomization (20 weeks of age) and feeding leucine to the animals either prevented the development of (primary prevention) or even reversed (secondary prevention) diastolic dysfunction. 

A significantly higher body weight was observed in HFpEF, HFpEF-prim, and HFpEF-sec compared to Con, with no difference between the HFpEF groups. As expected, a higher heart weight and a lower lung wet/dry ratio were seen between the HFpEF animals and the Con with no difference between the HFpEF animals (Table 1). No difference between the groups was obvious for tibia length. SO weight was higher in the HFpEF and HFpEF-sec group when compared to Con, whereas no difference was detected in the SO cross-sectional area (CSA) between all groups (Table 1). 

### 3.2. Impact of Leucine on Muscle Trophicity and Function

Upon measuring the SO and extensor digitorum longus (EDL) muscle weight and SO cross-sectional area (CSA) at an age of 32 weeks, no significant differences in muscle weight and SO CSA were evident between the three HFpEF groups (Table 1). Comparing the SO and EDL muscle weight of the HFpEF groups to the Con, a higher muscle weight was evident (Table 1). 

Analyzing the skeletal muscle function of the soleus muscle ex vivo in an organ-bath setting revealed a significantly enhanced absolute (Figure 2A,B) and muscle-specific force (Figure 2C,D) in HFpEF-prim compared to HFpEF. At least for the absolute force generation, the value of the HFpEF-prim group was not different compared to the control group. With respect to muscle fatigue, all the HFpEF animals showed a significantly lower fatigue level compared to the control animals. No impact of leucine treatment with respect to muscle fatigue was evident (Figure 2E).

### 3.3. Mitochondrial Function

Using the saponin-skinned muscle fiber technology, mitochondrial function was assessed in the soleus muscle of all three HFpEF groups and the control group. The assessment of complex I state 3 (addition of ADP and glutamate/malate or glutamate/malate/octanoylcarnitin as substrate) showed no difference between the groups (Figure 3A,B). After the addition of succinate and rotenone (state 3 complex II), a significantly reduced respiration was evident in the soleus muscle of the HFpEF group when compared to the control group. This significant reduction in the respiration was no longer evident after primary prevention with leucine whereas the secondary prevention had no effect when compared to untreated HFpEF (Figure 3C). Finally, the maximal respiration rate was measured for complex IV and demonstrated a significantly reduced respiration rate in all three HFpEF groups when compared to the control group (Figure 3D).

To investigate if alterations in mitochondrial respiration were related to changes in overall mitochondrial content or different expression levels of the respiratory chain complexes, Western blot analysis and enzyme activity measurements were performed. For the assessment of mitochondrial content, citrate synthase activity (Figure 4A) and the protein expression of porin (VDAC) (Figure 4B) were quantified. No differences for both markers were noticed between all groups. Regarding protein expression of the respiratory chain complexes I-IV (Figure 4C–F), a small but significant upregulation was detected for complex II and III in HFpEF-sec compared to HFpEF. Protein expression levels of complex I (Figure 4C) and IV (Figure 4F) were comparable between all groups. 

### 3.4. Proteins Related to Mitochondrial Function

Western blot analyses were performed to quantify the protein expression of proteins involved in modulating mitochondria, like Fis1 (Fission, Mitochondrial 1), mitofilin, mitochondrial creatine kinase (mi-CK), UCP3 (uncoupling protein 3), and mitochondrial Ca^2+^ uniporter (MCU). The expressions of Fis1 and Mfn-2, proteins involved in mitochondrial fusion and fission, were significantly increased in the soleus muscle of HFpEF-prim and HFpEF-sec compared to untreated HFpEF (Figure 5A,B). Mitofilin (also known as Mic60) plays critical roles in cristae organization, protein transport, mitochondrial DNA transcription, and ATP generation. Analyzing the expression of mitofilin in the soleus muscle of the three groups showed a significant upregulation in HFpEF-prim and HFpEF-sec compared to HFpEF (Figure 5C). Mi-CK, an enzyme responsible for the transfer of high-energy phosphate from the mitochondria to the cytosol, was significantly upregulated in HFpEF-prim compared to HFpEF (Figure 5D). UCP3, a protein facilitating the transfer of protons from the outer to the inner mitochondrial membrane and thereby reducing the mitochondrial membrane potential, was significantly reduced in the soleus muscle of HFpEF-prim animals compared to HFpEF and HFpEF-sec animals (Figure 5E). No difference was noted between the three groups for MCU (Figure 5F), a transmembrane protein that allows the passage of calcium ions from a cell’s cytosol into the mitochondria. Representative Western blots for the proteins mentioned above are depicted in Figure 5G.

### 3.5. Proteins Related to Muscle Atrophy

To assess molecular pathways related to muscle mass regulation, the protein expression of selected proteins was quantified by Western blot analyses. No difference was detected for MuRF1 (Figure 6A) and MAFBx (Figure 6B) protein expression in the soleus between all groups.

## 4. Discussion

Exercise intolerance is a hallmark of patients with HF independent of HFrEF (HF with reduced ejection fraction) or HFpEF. Experimental models of HF revealed functional and molecular alterations in the skeletal muscle ranging from muscle dysfunction to elevated fatigability to the induction of muscle atrophy [41,42]. The best-proven therapeutic approach targeting these alterations is exercise training [43,44,45]. Especially for HFpEF, other therapeutic approaches are, with the exception of Empagliflozin [9,25], rare and not very well investigated. In the present study, we evaluated the impact of leucine supplementation (primary and secondary prevention approach) on skeletal muscle performance in a validated animal model of HFpEF, since it is described that leucine modulates muscle atrophy and dysfunction [32,36]. The results of the present study can be summarized as follows: leucine supplementation, as a primary preventive approach,

prevents the development of muscle dysfunction (soleus absolute and specific muscle force) without modulation of muscle fatigability;improves mitochondrial function (mainly complex II induced respiration), without modulating mitochondrial content and the protein expression of respiratory chain complexes;increases the expression of mitofilin, mi-CK and reduces the protein expression of UCP3.

In summary, these results suggest that leucine supplementation, as a primary preventive approach, has the potential to prevent the development of skeletal muscle dysfunction, probably by targeting the mitochondria. Treatment of already established HFpEF with leucine (secondary prevention approach) seems to be ineffective regarding a functional improvement of the skeletal muscle, despite improved myocardial function [35]. For a summary of the results see Figure 7.

### 4.1. Leucine Supplementation and Skeletal Muscle Function

The most effective way to counteract loss of skeletal mass and function in HF is exercise training [46,47]. Unfortunately, older and very cachectic people are often unable to perform exercise training, and therefore nutritional interventions remain the most promising measure to delay loss of muscle mass and function. Beneficial effects of leucine supplementation on sarcopenia and muscle strength especially in elderly people have been reported (for review see [48,49]), but positive effects were also seen in middle-aged adults during prolonged bed rest [29]. With respect to muscular dysfunction in HFpEF and the impact of leucine supplementation, the present study is the first to investigate this important topic. Using the ZSF1 rat, an established animal model of HFpEF [8,9,37,38], the supplementation with leucine prevented the development of muscle dysfunction as documented by improved muscle force generation of the soleus muscle (primary preventive approach). These positive effects on muscle strength were independent of changes in muscle mass, since the mass and the cross-sectional area of the soleus muscle were not altered by leucine supplementation. This is in accordance with a recent study of leucine supplementation in an animal model of tumor cachexia [32] (also a primary prevention model since leucine was given at the time of tumor cell inoculation) where an improvement was seen in muscle strength without changes in the myofiber cross-sectional area. Also, in a bed rest study, leucine supplementation partially protected muscle function (measured as VO_2peak_) without having an impact on lean muscle mass [29]. The observation that leucine supplementation has no impact on muscle mass goes along with the fact that we did not detect an effect of leucine supplementation on the muscular expression of MuRF1 and MAFBx, two important proteins regulating muscle mass [50,51]. Altogether, this implies that leucine exerts positive effects on muscle strength by modulating metabolic pathways or other structural components important for muscle strength (see discussion below). Unfortunately, no effect on muscle strength was observed in animals treated with leucine after the onset of HFpEF and muscle dysfunction (secondary prevention approach). Therefore, leucine might only prevent the initiation of the functional loss, but is ineffective in reversing the negative impact of HFpEF on skeletal muscle function. This finding may be clinically important for the timing to start a potential nutritional intervention in the setting of HFpEF. Even in the research field investigating dietary strategies to manage sarcopenia, the timing of intervention is still under debate, and the research in this area focuses more on identifying patients at risk to start primary prevention strategies [52].

### 4.2. Leucine Supplementation and Mitochondrial Function

In numerous publications, it is well documented that the development of HFpEF and the associated skeletal muscle dysfunction are linked to modulations in mitochondrial function, content, and/or biogenesis [17,53]. The maintenance of mitochondrial “health” is essential for optimal energy production and proper functioning of specific tissues like the skeletal muscle and heart. The results of the present study clearly showed that leucine supplementation in the setting of primary prevention had a positive impact on mitochondrial function (mainly on complex II respiratory function) and proteins related to mitochondrial structure/function (mitofilin and UCP3), mitochondrial dynamics (Fis-1, MFN-2), or energy transfer (mi-CK). Of note, these changes in mitochondrial function and specific protein expression were not due to a general increase in mitochondrial content, since the enzyme activity of citrate synthase and the expression of porin (both accepted markers for assessment of mitochondrial content) were not different between the three groups. With respect to mitochondrial respiratory chain complex expression, even a significant increase was noted for complex II and III in HFpEF-sec, despite no change in respiratory function when compared to untreated HFpEF. This may be interpreted as a compensatory effect of the soleus muscle in HFpEF-sec animals to increase the energy supply for proper muscle function. 

The impact of leucine supplementation on mitochondrial function is in accordance with reports of the current literature [36,54,55]. In the study by Arentson-Lantz [55], healthy, older adults were supplemented with leucine or alanine (as control group) during 7 days of bed rest and 5 days of inpatient rehabilitation. The authors reported an increased oxygen consumption of complex II (after the addition of succinate) in the leucine-treated group, again with no changes in the protein expression of the respiratory chain complexes. Also, in an experimental study during cancer cachexia, leucine had a positive influence on mitochondria as determined by metabolomic and proteomic profiling [36].

Besides the modulation of mitochondrial respiratory function, leucine supplementation also influenced the expression of proteins related to mitochondrial structure, function, and energy transfer. Uncoupling protein 3 (UCP3) was significantly lower in the soleus muscle of HFpEF-prim compared to the other two groups. UCP3 is a member of the uncoupling protein family (UCP1, UCP2, and UCP3). UCP1 is mainly expressed in brown adipose tissue, whereas UCP2 is the main isoform in the myocardium and UCP3 in the skeletal muscle (for review see [56]). The exact physiological role of UCP2 and UCP3 is still discussed controversially. Already in 1997, Gong and colleagues showed that UCP3 overexpression lowered the proton gradient across the inner mitochondrial membrane and thereby modulated oxidative phosphorylation [57]. This modulation of oxidative phosphorylation by UCP3 was recently supported by an in vivo analysis [58]. Animals overexpressing UCP3 in their skeletal muscle exhibited mitochondrial inefficiency in vivo as documented by a 42% reduction in ATP synthesis. Therefore, one may assume that the reduction in UCP3 in HFpEF-prim is one possible way to improve mitochondrial function and the efficiency of ATP production without modulating the protein expression of respiratory chain complexes. Another enzyme important for energy transfer from the mitochondria to the place of utilization (myofibril ATPase) is creatine kinase (CK) and especially mi-CK. The mi-CK is located in the mitochondrial intermembrane space where it utilizes the ATP generated by the mitochondria to phosphorylate creatine, which is then transported to the myofibril where it transfers the energy to the myofibril ATPase for contraction [59]. The observed increase in mi-CK expression in HFpEF-prim has to be interpreted together with the increased mitochondrial function in this group. In summary, it seems that leucine, as a primary preventive approach, increased mitochondrial function and also the efficiency of transporting the energy to the contractile machinery. This modulation of the mi-CK system by exercise training has already been documented in skeletal muscle biopsies of volunteers [60]. In addition, leucine supplementation (independent of primary or secondary approach) had an impact on Fis1 and mitofilin, both important proteins regulating mitochondrial dynamics. Mitofilin (also known as Mic60), originally described as a heart muscle protein with high expression in this tissue [61], has been described as the main subunit of the MICOS (mitochondrial contact site and cristae organizing system) complex and is important for the proper formation of the cristae structure in mitochondria. The downregulation of mitofilin resulted in the instability of the MICOS components, the disassembly of the MICOS complex, and a disorganized cristae structure [62]. Furthermore, the overexpression of mitofilin alleviated acute myocardial infarction, triggered inflammation in the heart, and reduced pyroptosis-related factors [63]. Therefore, the upregulation of mitofilin by leucine supplementation can be regarded as a protective mechanism to prevent the loss of mitochondrial function. Besides mitochondrial organization, mitochondrial fusion and fission also seem to be altered by leucine treatment. In primary as well as secondary prevention, a significant upregulation of FIS-1 and MFN-2 was observed. The leucine-induced upregulation of MFN-2 may be regarded as a mechanism to preserve or increase mitochondrial function because the knockout of MFN-2 induces mitochondrial respiratory dysfunction [64].

Nevertheless, the sole modulation of mitofilin and mitochondrial fusion and fission is not enough to prevent mitochondrial dysfunction, since the mitochondrial and skeletal muscle function in secondary prevention did not improve despite the increased expression of mitofilin. Further experiments are therefore warranted to dissect the difference between leucine in primary and in secondary prevention with respect to the improvement in skeletal muscle function in HFpEF. 

### 4.3. Study Limitations 

Despite the encouraging effects of leucine supplementation in the setting of primary prevention on the modulation of skeletal muscle alterations in HFpEF, some limitations have to be mentioned.

First, in the present study, we did not measure exercise performance (running test or cardiopulmonary exercise test) directly. This might be of relevance since it is described in the current literature that, besides alterations in the skeletal muscle, other factors like the capillarization of the muscle and the ability of the myocytes to extract oxygen from the blood contribute to exercise intolerance [7,11,65].

Second, no histological analyses such as electron microscopy were performed. Electron microscopy would provide critical information about mitochondrial content, cristae density, organization, and any other abnormalities commonly observed in heart failure. Recently, we demonstrated the accumulation of intermyofibrillar and subsarcolemmal mitochondria, with a predominately small size, in the diaphragm of heart failure patients [66].

Third, only female rats were used in the present study. Therefore, the transferability of the present results to the total HFpEF population needs to be confirmed. Nevertheless, looking at data from registry- and community-based studies, it is clear that the majority of patients with HFpEF are older, female, and have multiple comorbidities, including hypertension, diabetes, pulmonary disease, chronic kidney disease, and obesity [67,68].

Fourth, the functional and molecular analyses were only performed in the soleus muscle, representative of a type I skeletal muscle. Since mitochondrial function was an important readout in the present study, we selected the soleus muscle, due to its higher mitochondrial content in comparison to the EDL muscle. Nevertheless, more investigations are necessary in other muscle groups to generalize the results shown in the present study. 

## 5. Conclusions

In conclusion, the results of the present study show for the first time that leucine supplementation, at least as a primary prevention approach, has the capability to prevent skeletal muscle functional loss in HFpEF. It seems that the positive effects on muscle function are partially mediated by improving mitochondrial function through modulating energy transfer and mitochondrial structure and coupling. In the setting of secondary prevention, leucine had no effect on skeletal muscle function.

## Figures and Tables

**Figure 1 cells-13-00502-f001:**
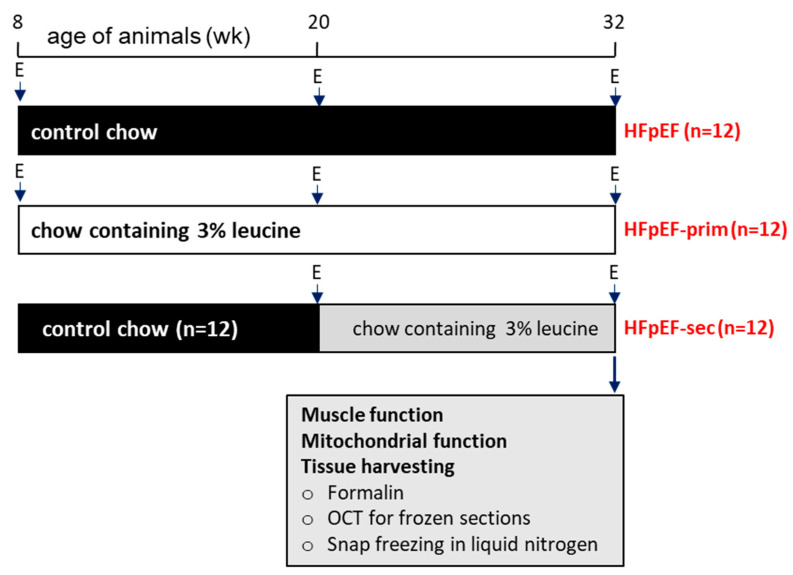
Study design. A total number of 36 ZSF1 obese rats were included. After baseline echocardiography, the animals were randomized into three groups either receiving no intervention (HFpEF, n = 12) or leucine (standard chow enriched by 3% leucine) for 24 weeks (HFpEF-prim, n = 12) or for 12 weeks (HFpEF-sec, n = 12). Final echocardiography was performed at 32 weeks of age prior to organ harvest. The soleus and EDL muscles were collected for histological and molecular analyses. E = echocardiography.

**Figure 2 cells-13-00502-f002:**
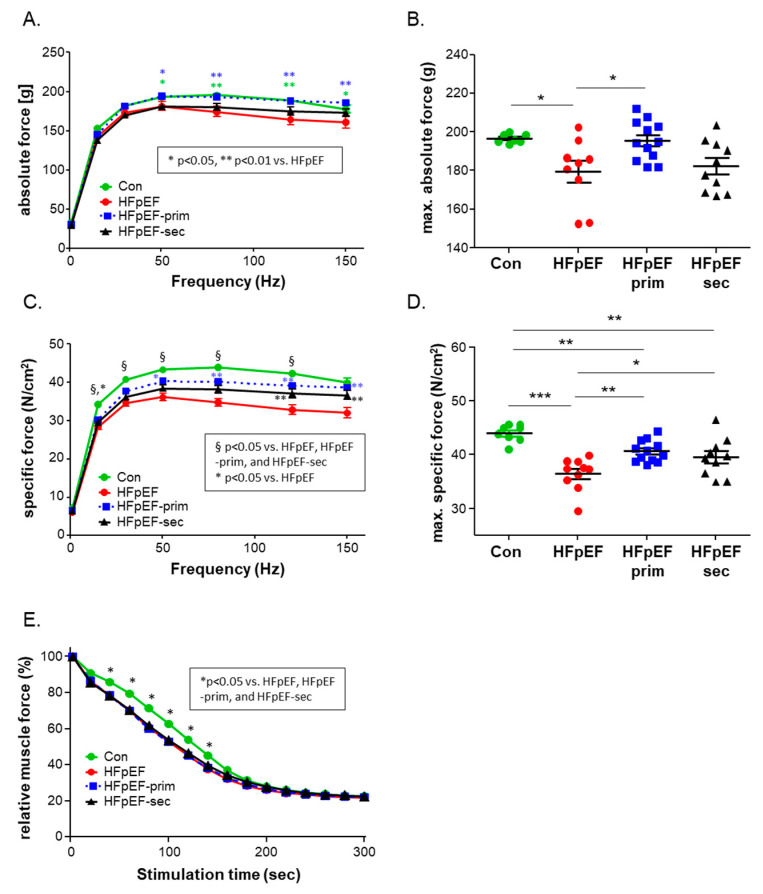
Effect of leucine supplementation on soleus muscle function at 32 weeks of age. Soleus absolute (**A**) and maximal absolute force (**B**) were significantly decreased in HFpEF (red circles) when compared to control (green circles). Treatment with leucine in primary prevention (blue squares) resulted in a significantly higher muscle force when compared to HFpEF. This increase was not seen in the HFpEF-sec group (black triangles). The specific force (**C**) and the absolute specific force (**D**) were significantly decreased in all HFpEF groups compared to control. A significant increase was seen after leucine treatment (HFpEF-prim) when compared to the HFpEF group. Furthermore, soleus muscle fatigue was assessed by repetitive stimuli in all groups and expressed as relative muscle force (**E**). * *p* < 0.05, ** *p* < 0.005; *** *p* < 0.001.

**Figure 3 cells-13-00502-f003:**
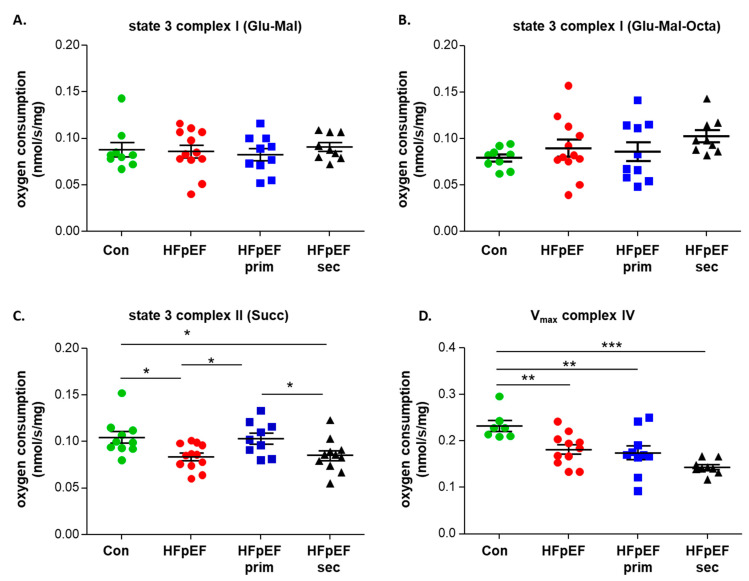
Effect of leucine supplementation on mitochondrial respiratory function of HFpEF animals. Stimulation of complex I with glutamate/malate (Glut/Mal) (**A**) or Glut/Mal/octanoylcarnitin (Octa) (**B**) was comparable between groups. Furthermore, stimulation of complex II with succinate (**C**) revealed a significant increase in oxygen consumption in HFpEF-prim when compared to HFpEF and HFpEF-sec. To measure the maximal respiration of complex IV (**D**), ascorbate (Asco) and the electron-donating compound, tetramethyl-p-phenylene diamine (TMPD), were added after the addition of the mitochondrial uncoupler FCCP. No significant differences were observed between the three HFpEF groups, which all showed a lower respiration rate compared to the control group. * *p* < 0.05, ** *p* < 0.01, *** *p* < 0.001.

**Figure 4 cells-13-00502-f004:**
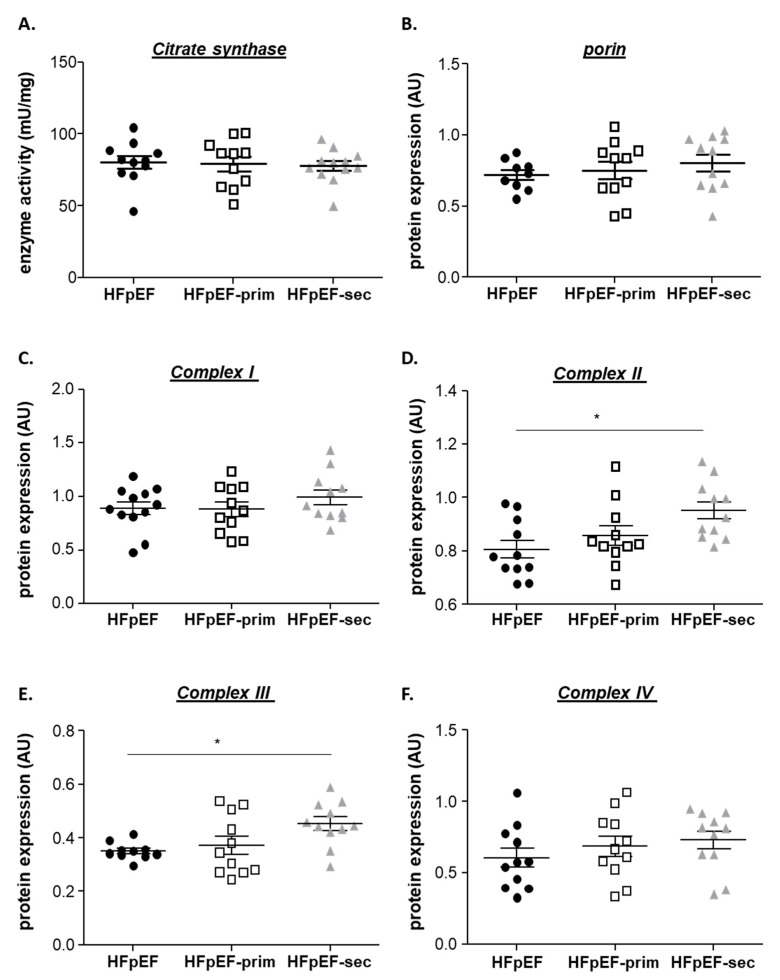
Effect of leucine supplementation on enzyme activity and mitochondrial content. Enzyme activity of citrate synthase (**A**) was comparable between groups, as well as protein expression of porin (**B**) and mitochondrial respiratory chain complex I (**C**). Protein expressions of complex II (**D**) and III (**E**) were increased in HFpEF-sec when compared to HFpEF. No significant difference was observed in protein expression of complex IV (**F**). * *p* < 0.05.

**Figure 5 cells-13-00502-f005:**
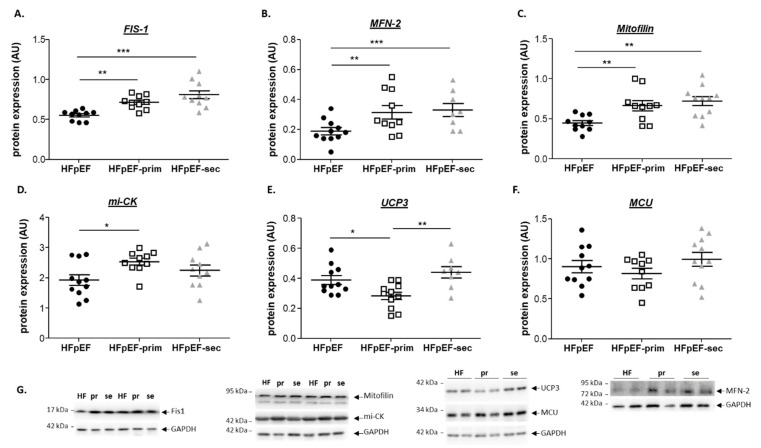
Effect of leucine supplementation on proteins related to mitochondrial function. Protein expressions of mitochondrial fission protein 1 (Fis1) (**A**) MFN-2 (**B**) and mitofilin (**C**) were increased in both prevention approaches when compared to HFpEF. Mitochondrial creatine kinase (mi-CK, (**D**)) protein expression was increased in HFpEF-prim but not in HFpEF-sec when compared to HFpEF. Uncoupling Protein 3 (UCP3, (**E**)) protein expression was downregulated in HFpEF-prim when compared to HFpEF and HFpEF-sec. No significant difference was observed in protein expression of mitochondrial calcium uniporter (MCU, (**F**)). Representative Western blot bands are depicted (**G**). * *p* < 0.05, ** *p* < 0.005; *** *p* < 0.001.

**Figure 6 cells-13-00502-f006:**
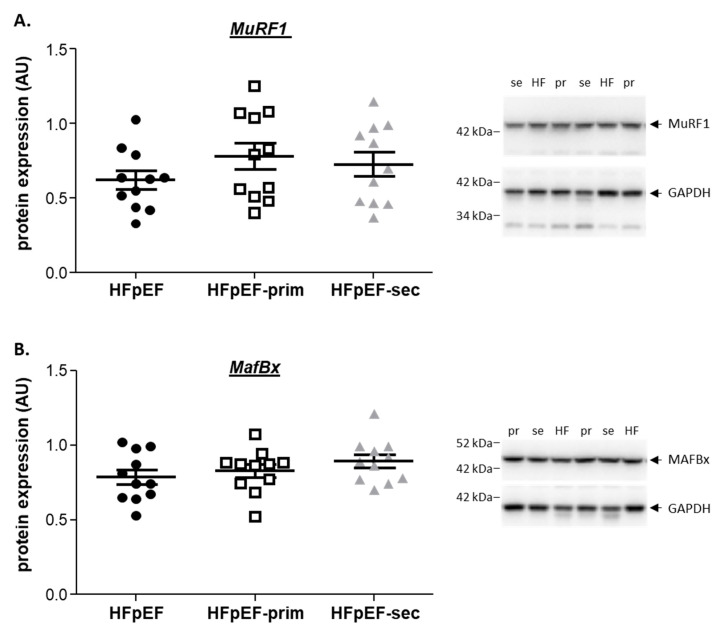
Impact of leucine supplementation on proteins related to muscle atrophy. Densitometric analysis of MuRF1 (**A**) and MafBx (**B**) protein expression revealed no differences between the three groups.

**Figure 7 cells-13-00502-f007:**
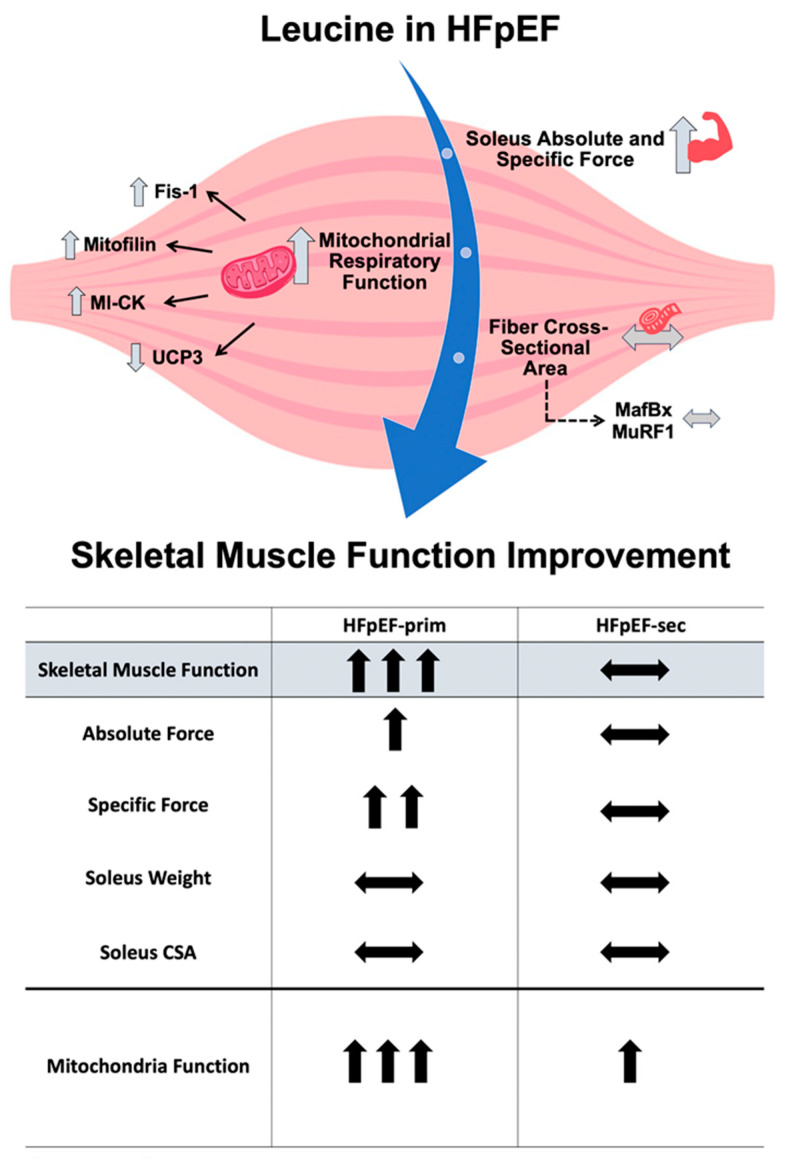
A summary of the beneficial impacts of leucine supplementation on skeletal muscle dysfunction as observed in HFpEF. ↑ increased vs. HFpEF; 
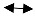
 no change vs. HfPEF.

**Table 1 cells-13-00502-t001:** Animal characteristics at an age of 32 weeks.

	Con	HFpEF	HFpEF-prim	HFpEF-sec
Body weight (g)	260 ± 3	512 ± 7 ***	500 ± 7 ***	486 ± 5 *
Tibia length (mm)	35.3 ± 0.1	35.5 ± 0.1	35.5 ± 0.1	35.4 ± 0.2
Heart weight/TL (mm/mg)	26.5 ± 0.4	37.8 ± 0.7 ***	37.3 ± 0.5 ***	37.7 ± 1.0 **
Lung weight (wet/dry)	4.46 ± 0.02	4.14 ± 0.03 ***	4.15 ± 0.02 ***	4.20 ± 0.03 ***
Soleus weight/TL (mg/mm)	4.00 ± 0.04	4.42 ± 0.11 **	4.27 ± 0.06	4.39 ± 0.10 *
Soleus CSA (µm^2^)	3003 ± 162	3112 ± 118	2908 ± 150	2678 ± 79
EDL weight/TL (mg/mm)	3.78 ± 0.06	4.07 ± 0.04 **	4.07 ± 0.02 **	4.04 ± 0.06 *

TL = tibia length; CSA = cross-sectional area; EDL = extensor digitorum longus; * *p* < 0.05, ** *p* < 0.01, *** *p* < 0.001 vs. Con.

## Data Availability

The data that support the findings of the study are available from the corresponding author upon reasonable request.

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
