# Peer review of "Leucine Supplementation Prevents the Development of Skeletal Muscle Dysfunction in a Rat Model of HFpEF"

_cells, 2024, doi:10.3390/cells13060502_

Round 1

Reviewer 1 Report

Comments and Suggestions for Authors

In summary, this is a well-designed study demonstrating the benefits of leucine supplementation on skeletal muscle in an animal model of HFpEF. Some limitations exist, and further analyses would strengthen the conclusions. However, the study advances our mechanistic understanding of nutritional approaches to combat exercise intolerance in HFpEF.

Major comments:

A key finding of your study is that leucine supplementation prevented muscle dysfunction when given as primary prevention but did not reverse existing dysfunction when given as secondary prevention. This difference in efficacy based on the timing of intervention should be highlighted and discussed further, as it has important clinical implications regarding the potential window for nutritional interventions. Please expand the discussion addressing why secondary prevention failed to improve muscle function despite some mitochondrial changes observed.

While the extensive mitochondrial respiration assays are a strength, the mechanistic link specifically tying improved respiration to enhanced muscle contractility with leucine supplementation needs to be explored further. For instance, is this due to increased ATP supply or changes in calcium handling or myofilament structure? Elucidating these causal pathways could strengthen conclusions.

The exclusive use of female rats, while consistent with the predominantly female HFpEF demographic, may limit the generalizability of the findings to the broader HFpEF population. The authors should acknowledge this as a limitation and discuss any particular reasons for excluding male rats from the study design.

Minor comments:

Additional histological or electron microscopic analyses may have provided complementary ultrastructural information on muscle morphology changes underlying the functional improvements observed with leucine.

Reviewer 2 Report

Comments and Suggestions for Authors

The study by Nascimento Alves et al describes therapeutic effects of leucine in a model of heart failure and focusses on skeletal muscle health. There are some concerns regarding this study that need to be addressed.

Major remarks

My main concern is that the research choices that were made in this study are at times debatable, and have not been justified properly. Firstly, the choice for soleus needs more information. Is this choice made based upon higher mitochondrial content? What about representativity to muscle phenotype, and is there an indication that differences between muscle groups could be present?  Secondly, the choice of mitochondrial markers shows a strong focus on mitochondrial dynamics. Yet, the spectrum of mitochondrial fission-fusion is not represented, as MFN is lacking from the analyses. This needs to be addressed and remedied. Also explain in more detail the importance of mitochondrial morphology and dynamics in muscle dysfunction and provide more context in relation to hearth failure. How does this compare to cardiomyocyte dysfunction? Thirdly, choices for starting therapies and leucine dose need explanations.  Add to the text at what ages first heart and skeletal muscle symptoms occur in this model.

The study lacks a healthy control group, which makes it difficult to evaluate the relevance of the observed beneficial effects on muscle force. It is understandable that no controls are included in the treatment groups, but it would have been possible to provide the limited data comprised in Table 1 for a control group. This point should be added to the study limitations section. Is the effect meaningful in view of what can be expected in healthy age-matched rats? What % of normal value represents the described improvement of force? Also, CSA of fibers is unchanged, but what about in comparison to healthy rats? Is CSA reduced in the model? If so, this means that this would persists with leucine treatment.

The study is purely descriptive, yet allegations are made that are not supported by the presented results. Ex. lines 336, 433.  I would advise to also change the title and the last sentence of the abstract to better fit the study .  

Full blots for Figs 5 and 6 need to be provided as a supplement.

Minor remarks

Line 186: correct E/é

Comments on the Quality of English Language

Quality of English is good.

Round 2

Reviewer 1 Report

Comments and Suggestions for Authors

I don't have any further comments to add.

Author Response

Thank you very much and we appreciated all your comments.

Reviewer 2 Report

Comments and Suggestions for Authors

The authors have adequately addressed the concerns raised in the revised manuscript. The full blot for MFN2 however remains missing.

Comments on the Quality of English Language

English language is fine

Author Response

We have to apologize, but during the first revision we forgot to add the full blot for MFN2 in the supplementary material. We added the full blot now into the supplementary material full western blots.